# An Estimated $\delta$-Based Iterative Block Decision Feedback Equalization in SC-FDE System

**Yidong Liu *** , **Xihong Chen, Dizhe Yuan and Denghua Hu**

Air Defense and Antimissile School, Air Force Engineering University, Xi'an 710051, China
* Correspondence: lyd19981205@163.com

**Abstract:** We provide a novel nonlinear frequency domain equalization algorithm for the frequency domain equalization of an SC-FDE system by improving the classical iterative block decision feedback equalization (IBDFE) algorithm and applying $\delta$ estimation to the improved algorithm. The improvement of the IBDFE algorithm is carried out by replacing the ZF equalization in the feedback branch with the MMSE equalization and eliminating the iteration of the correlation factor, thus reducing the noise error and the computational complexity of the original algorithm. $\delta$ estimation can estimate residual inter-symbol interference in the signal after MMSE equalization and reject it, thus further improving the equalization accuracy. The simulation results show that the performance of the novel algorithm is better than that of the IBDFE algorithm with similar complexity, or the complexity of the novel algorithm is lower than that of the IBDFE algorithm with similar performance.

**Keywords:** single-carrier frequency domain equalization; iterative block decision feedback equalization; $\delta$ estimation





## 1. Introduction

The multipath fading problem in the process of signal transmission is one of the main factors restricting the development of high-speed wireless communication. It manifests as inter-symbol interference (ISI) in the time domain, which leads to signal distortion, thus affecting the reliability and stability of the system [1,2]. Orthogonal frequency division multiplexing (OFDM) technology is an effective means to combat the multipath problem, but OFDM technology has a high Peak to Average Power Ratio (PAPR), which requires the range of the linear region of the power amplifier to be large, increasing the system cost and a decrease in signal quality [3–5]. The single-carrier frequency domain equalization (SC-FDE) [6] technology, which was proposed by H. Sari et al. in the 1990s, can overcome the shortcomings of the above-mentioned OFDM technology and has equivalent transmission capacity and complexity. Therefore, SC-FDE is adopted by the IEEE802.ad and IEEE802.11ay standards as the physical layer transmission scheme therein. The design of a frequency domain equalizer is one of the key technologies of the SC-FDE system [7,8].

In SC-FDE systems, the time-varying and multipath characteristics of the channel can cause signal distortion and thus ISI, which can lead to a high bit error rate (BER) during symbol detection if the errors are not effectively compensated at the receiver. Therefore, for the SC-FDE wireless communication system, the channel state information (CSI) is obtained after the channel estimation and the necessary frequency domain equalization methods must be adopted to compensate for the channel effects. Common frequency domain equalization algorithms can be divided into two categories: linear equalization and nonlinear equalization. The equalization in which the decision output is not used for feedback is called linear equalization [9,10], such as the classic zero-force equalization (ZF) and minimum mean square error (MMSE) equalization. ZF equalization directly uses the inverse matrix of the channel impulse response matrix as the filter coefficient, which is small in computation and low in complexity. However, when the channel has deep fading

poles, the noise increases, and the equalization performance decreases [11]. The purpose of MMSE is to optimize the mean square error to the minimum. It is less affected by noise at the deep fading pole of the channel, and its performance is better than that of ZF, but it has a certain residual inter-symbol interference (RISI) [12]. The linear equalization algorithm has low complexity but has limitations. Therefore, it is necessary to investigate nonlinear equalization algorithms with feedback mechanisms to achieve more accurate and efficient frequency domain equalization.

This paper studies the classical IBDFE algorithm, adopts the basic idea of estimating the signal for feedback decision, redesigns the feedback branch for the problems of high computational complexity and the large error of ZF equalization in the classical IBDFE algorithm, and uses MMSE equalization instead of ZF equalization to estimate the transmitted signal. Then considering the existence of RISI in MMSE equalization, this paper applies $\delta$ estimation to remove RISI in MMSE equalization and proposes an estimated $\delta$-based iterative block decision feedback equalization (E-IBDFE) algorithm to further improve the system equalization performance.

The rest of this paper is organized as follows. In Section 2, we present an overview of the related work. Section 3 discusses the characteristics of the SC-FDE system. In Section 4, the estimated $\delta$-based iterative block decision feedback equalization is introduced. In Section 5, several examples are described. Some conclusions are drawn in Section 6.

## 2. Related Work

A lot of research work has been conducted on nonlinear frequency domain equalization algorithms by scholars from different countries. QI Y L [13] proposed joint channel estimation and equalization with frame structure based on UW, which uses noise prediction and removes it in channel estimation to derive a more accurate channel estimate; on the other hand, the more accurate channel estimate is substituted into the equalization algorithm to improve the accuracy of frequency domain equalization. Salman M B [14] proposed a receiver structure that combines the outputs of the fractional delayed bank of FDEs to overcome the performance degradation of FDE for highly frequency selective channels with nonlinear distortion. XIE Z D [15] studied a joint channel estimation and equalization algorithm over time-frequency doubly selective channels, which enhances the information interaction between the two and achieves the joint optimization of channel estimation and equalization. BAI G [16] designed a subnetwork for each of the three modules, channel estimation, noise power estimation, and channel equalization, and applied deep learning to the SC-FDE algorithm to reduce the amount of training data required for network convergence and improve the channel generalization capability. Cao T N [17] et al. considered diffusive molecular communication (MC) systems affected by signal-dependent diffusive noise, inter-symbol interference, and external noise, and designed linear and nonlinear fractionally-spaced equalization schemes and a detection scheme that combines decision feedback and sequence detection (DFSD). From the current research results, a considerable number of nonlinear frequency domain equalization algorithms have been proposed, but some of them are highly complex and computationally intensive, and still need further optimization and improvement.

## 3. System Model

The principle block diagram of the SC-FDE system is shown in Figure 1. Unlike the OFDM system, signal processing focuses on the transmitting end, and modulation and decisions are completed in the frequency domain. The signal processing of the SC-FDE system focuses more on the receiving end [3].

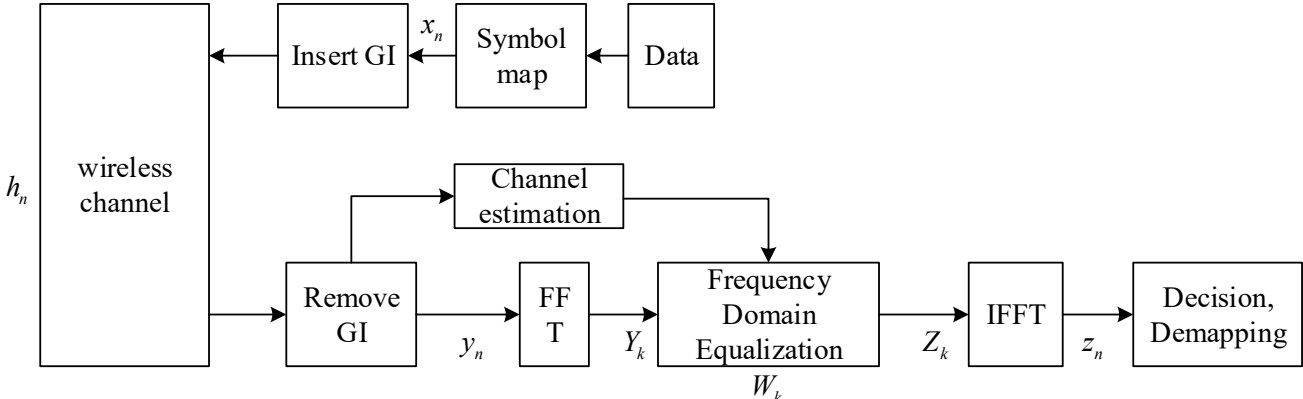

**Figure 1.** SC-FDE system principle structure diagram.

In the SC-FDE system, the transmitted signal first goes through symbol mapping, then inserts a cyclic prefix as a guard interval (GI), and then transmits it to the receiver through a wireless channel. After receiving the signal data, the receiving end first removes the GI, and then transfers the signal to the frequency domain through fast Fourier transformation (FFT) for channel estimation and frequency domain equalization operations, and, finally, through the inverse fast Fourier transformation (IFFT) converts the equalized signal data back to the time domain for the decision and demapping to obtain the final estimated signal.

In this paper, the SC-FDE system uses UW as the cyclic prefix insertion, and the system data frame structure after UW insertion is shown in Figure 2. This structure can effectively reduce the inter-symbol interference (ISI) caused by multipath effects. To avoid ISI, the UW length should be greater than the maximum delay spread $\tau_m$ of the channel and have good periodic autocorrelation.

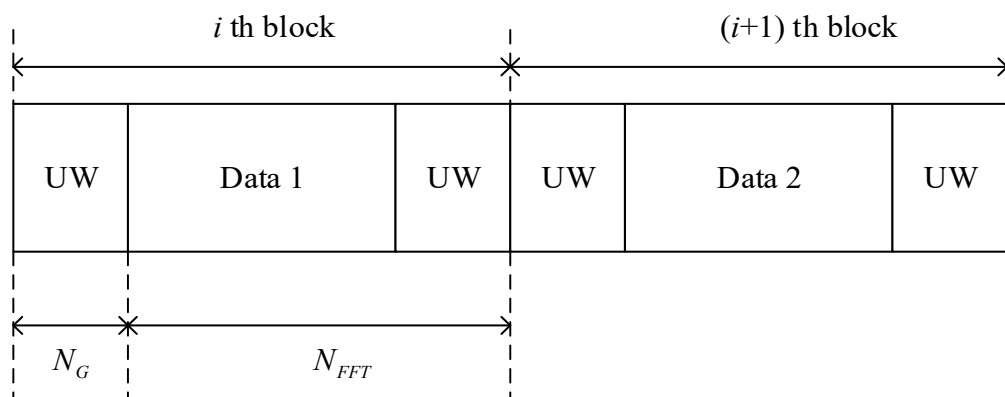

**Figure 2.** UW-based signal frame structure.

Assuming that the transmitted signal is $x_n = [x_0, x_1, \cdots, x_{N-1}]^T$, after inserting the GI, it is transmitted through the wireless channel, and the receiving end receives and removes the GI and the data are $y_n = [y_0, y_1, \cdots, y_{N-1}]^T$. According to reference [18], its time domain expression can be written as

$$y_n = h_n \otimes x_n + v_n \tag{1}$$

where $h_n$ is the channel impulse response matrix, and $v_n$ is the additive white Gaussian noise with mean 0 and variance $\sigma_\omega^2$.

After receiving the data through FFT, it can be expressed as

$$Y_k = F y_n = H_k X_k + V_k \tag{2}$$

$F$ is an $N \times N$-dimensional FFT matrix and the expression of the elements in the matrix is

$$[F]_{p,q} = \frac{1}{N} \exp\left( \frac{-j2\pi pq}{N} \right) \tag{3}$$

where $p, q = 0, 1, \cdots N - 1$.

## 4. Improved IBDFE Algorithm

### 4.1. IBDFE Algorithm

IBDFE is an equalization algorithm that gradually eliminates the influence of channel fading on signal amplitude and phase through multiple iterations. The basic principle of IBDFE is to estimate the correlation factor of the previous decision signal and the transmitted signal, to obtain new filter coefficients, and continue to iterate to approach the real value one step closer [19–21].

The schematic diagram of the principle of the IBDFE is shown in Figure 3.

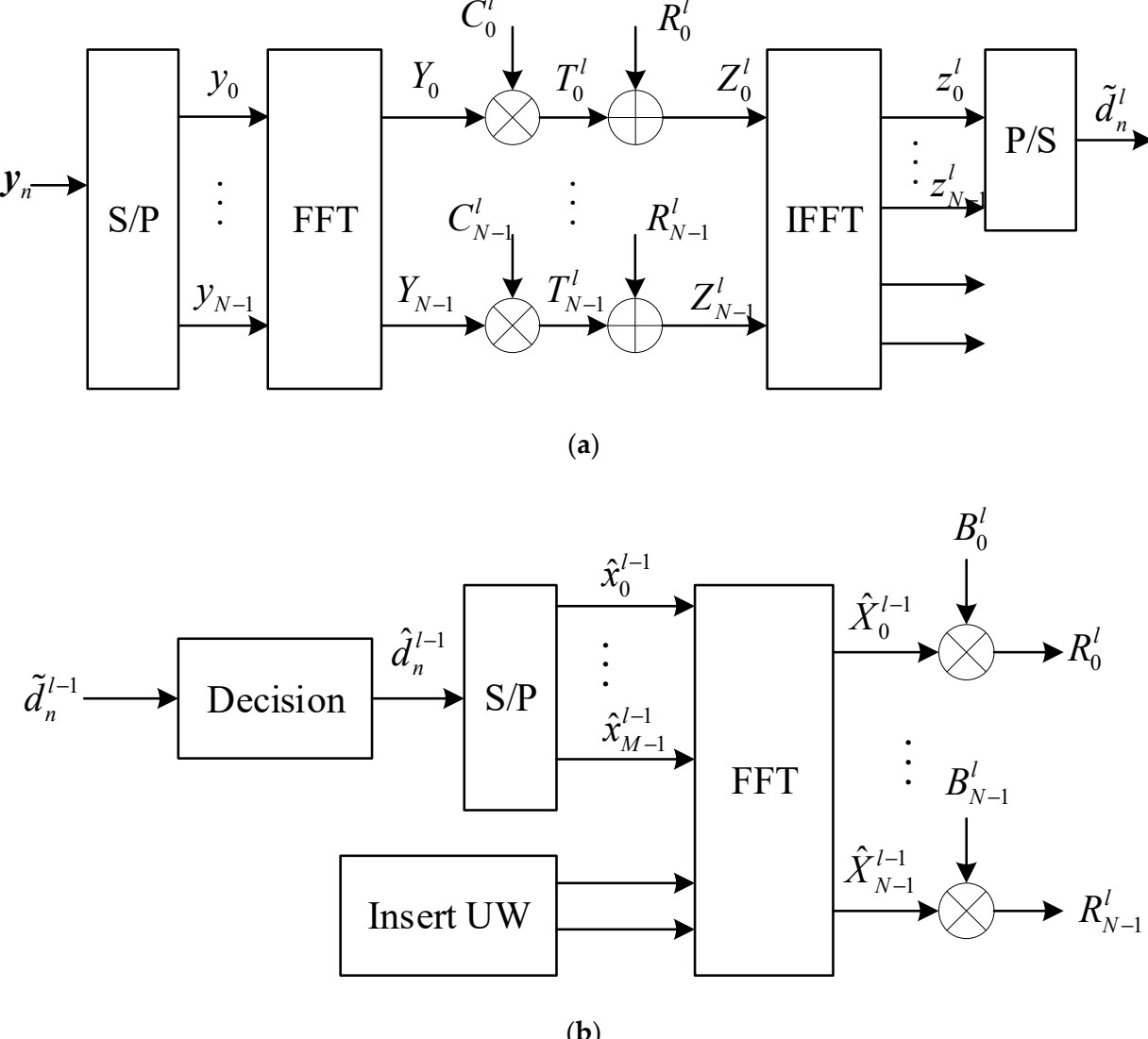

**Figure 3.** Schematic diagram of IBDFE. (**a**) Feedforward branch. (**b**) Feedback branch.

In Figure 3, $\left\{ B_k^l \right\}$ and $\left\{ C_k^l \right\}$ represent the feedback filter coefficients and feedforward filter coefficients, respectively, where $k = 0, 1, \cdots N - 1$, $l = 1, 2, \cdots N_l$, $l$ represents the number of iterations. In the feedforward branch, the received signal is multiplied by the

feedforward filter coefficients after serial–parallel conversion and FFT, and the resulting signal is denoted by T, that is

$$T_k^l = Y_k^l C_k^l \tag{4}$$

The signal sequence judged after the $(l-1)$th iteration is denoted by $\left\{ \hat{d}_n^{l-1} \right\}$, $\left\{ \hat{d}_n^{l-1} \right\}$ is obtained after serial-to-parallel conversion, UW insertion, and FFT to obtain $\left\{ \hat{X}_k^{l-1} \right\}$, which is multiplied by the feedback filter coefficient $\left\{ B_k^l \right\}$ to obtain the feedback branch output $\left\{ R_k^l \right\}$; $R_k^l$ can be represented as

$$R_k^l = \hat{X}_k^{l-1} B_k^l \tag{5}$$

Then the result of the $l$th iteration is

$$Z_k^l = T_k^l + R_k^l = Y_k^l C_k^l + \hat{X}_k^{l-1} B_k^l \tag{6}$$

After $Z_k^l$ is converted to the time domain by IFFT for the decision, the post-equalization information can be obtained.

Using the minimum mean square error criterion, if we define $J^l$ as the mean square error (MSE) of the detection point $\tilde{d}$, then we have

$$J^l = E\left[ \left| \hat{d}_n^l - x_n \right|^2 \right] = \frac{1}{N} \sum_{i=0}^{N-1} \left| Z_i^l - X_i \right|^2 \tag{7}$$

The power of the transmitted signal $\{x_n\}$ and the power of the estimated signal $\left\{ \hat{x}_n^l \right\}$ of the $l$-th iteration are, respectively, defined as

$$M_{X_k} = E\left[ |X_k|^2 \right] \tag{8}$$

$$M_{\hat{X}_k^l} = E\left[ \left| \hat{X}_k^l \right|^2 \right] \tag{9}$$

The correlation factor between the $M_{X_k}$ and $M_{\hat{X}_k^l}$ can be expressed as

$$r_{X_k, \hat{X}_k^l} = E\left[ X_k \hat{X}_k^l \right] \tag{10}$$

Substituting Equations (8)–(10) into Equation (7), we can obtain

$$J^l = \frac{1}{N} \sum_{k=0}^{N-1} \left\{ \left| C_k^l \right|^2 M_\omega + \left| C_k^l H_k - 1 \right|^2 M_{X_k} + \left| B_k^l \right|^2 M_{\hat{X}_k^l} + 2\mathrm{Re}\left[ B_k^{l*} \left( C_k^l H_k - 1 \right) r_{X_k, \hat{X}_k^l} \right] \right\} \tag{11}$$

where $M_\omega = N\sigma_\omega^2$ is the noise power in the frequency domain. The notation of $(\cdot)^*$ denotes the conjugation of the matrix, and $\mathrm{Re}[\cdot]$ denotes the real part.

According to reference [22], since the filter design cannot have an effect on the current signal, we can obtain

$$\sum_{k=0}^{K-1} B_k^l = 0 \tag{12}$$

If the result of taking the derivative of $J^l$ to $B_k^l$ and $C_k^l$, respectively is 0, we can obtain

$$B_k^l = -\frac{r_{X_k,\hat{X}_k^l}}{M_{\hat{X}_k^l}}\left[C_k^l H_k - \gamma^l\right] \tag{13}$$

$$C_k^l = \frac{{H_k}^*}{M_\omega + M_{X_k}\left(1 - \frac{\left|r_{X_k,\hat{X}_k^l}\right|^2}{M_{X_k}M_{\hat{X}_k^l}}\right)|H_k|^2} \tag{14}$$

where

$$\gamma^l = \frac{M_{X_k}\Phi^l}{1 + \frac{\left|r_{X_k,\hat{X}_k^l}\right|^2}{M_{\hat{X}_k^l}}\Phi^l} \tag{15}$$

$$\Phi^l = \frac{1}{K}\sum_{k=0}^{K-1}\frac{|H_k|^2}{M_\omega + \left(M_{X_k} - \frac{\left|r_{X_k,\hat{X}_k^l}\right|^2}{M_{\hat{X}_k^{l-1}}}\right)|H_k|^2} \tag{16}$$

The correlation factor can be obtained by the estimation method in reference [23]

$$r^l = \frac{1}{N_k}\sum_{k\in\Omega}\frac{Y_k}{H_k}\hat{X}_k^{l-1}{}^* \tag{17}$$

Considering the deep fading of the channel, if a threshold $H_{th}$ is set for $H_k$ in Equation (17), then we can obtain $\Omega = \{k : |H_k| \geq |H_{th}|\}$.

It can be seen from the above derivation process and the expression of the correlation quantity that the classical IBDFE algorithm needs to recalculate the correlation factor $r_{X_k,\hat{X}_k^l}$ of the transmitted signal $\{x_n\}$, and the estimated signal $\left\{\hat{x}_n^l\right\}$ iteratively updates the coefficient $C_k^l$ and the coefficient $B_k^l$ every time, resulting in a relatively large amount of calculation. Moreover, Equation (17) adopts the data after ZF equalization, namely $Y_k/H_k$, to estimate the transmitted signal $X_k$, which ignores the deviation between the estimated value and the actual value caused by signal noise. In addition, the accuracy of the correlation factor affects the performance of equalization. If $r^l$ is too large, ISI will increase; if $r^l$ is too small, it will cause slow convergence of iterative equalization, and the filter will have a poor effect on eliminating ISI.

### 4.2. Improved IBDFE Algorithm
#### 4.2.1. MMSE-IBDFE Algorithm

The main operations of the classical IBDFE algorithm are all performed in the frequency domain, which does not require matrix inversion, that reduces the computational complexity to a certain extent. However, the multiple iterations of the feedforward coefficients and feedback coefficients and the estimation of the correlation factors still result in a large amount of computation [24]. The estimation of the correlation factor A is calculated by using zero-forcing equalization to calculate the transmitted signal, but the ZF equalization will have a large error at the deep fading point of the channel [25]. In order to further reduce

the complexity of the algorithm and reduce the error caused by zero-forcing equalization, consider improving the feedback branch, and use the MMSE equalization instead of ZF equalization to estimate the transmitted signal.

The schematic block diagram of the feedback branch after the improvement is shown in Figure 4.

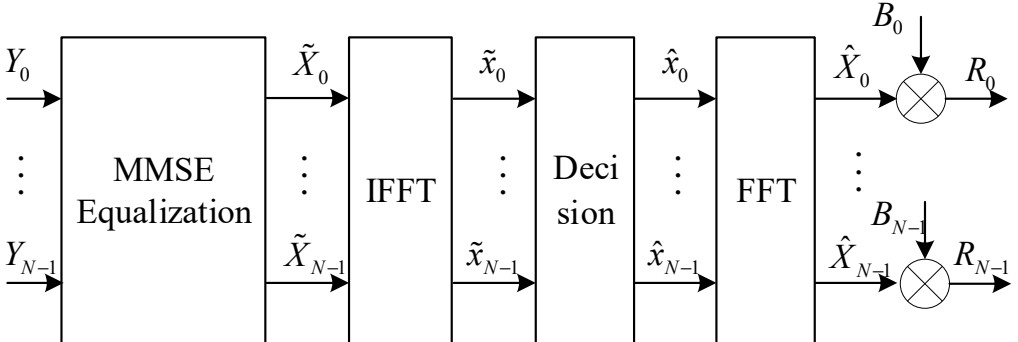

**Figure 4.** Feedback branch of MMSE-IBDFE.

It can be seen from Figure 4 that the received signal enters the feedback branch after serial–parallel conversion and FFT. After MMSE equalization and a decision, it is directly multiplied by the feedback filter coefficient $\{B_k\}$ to obtain $\{R_k\}$, thereby canceling the block iteration.

Substituting $\hat{X}_k$ for $\hat{X}_k^l$, the expression for $\{Z_k\}$ in the MMSE-IBDFE algorithm is obtained as

$$Z_k = Y_k C_k - \hat{X}_k B_k \tag{18}$$

According to Equation (2), combined with the MSE criterion, we can obtain

$$J_k = E\left[\left|Z_k - \hat{X}_k\right|^2\right] = E\left[|H_k C_k - B_k - 1|^2\left|\hat{X}_k\right|^2 + \\ |C_k|^2|V_k|^2 + 2\mathrm{Re}\left[(H_k C_k - B_k - 1)\hat{X}_k C_k V_k\right]\right] \tag{19}$$

Due to the correlation between the transmitted signal $\{X_k\}$ and the noise $\{V_k\}$, the above equation can be simplified as

$$J_k = E\left[\left|Z_k - \hat{X}_k\right|^2\right] = E\left[|H_k C_k - B_k - 1|^2\left|\hat{X}_k\right|^2 + |C_k|^2|V_k|^2\right] \tag{20}$$

In order to obtain the optimal solution, the Lagrangian function is defined as

$$f_\lambda = E\left[|H_k C_k - B_k - 1|^2 + |C_k|^2 \sigma_\omega^2 + \lambda \sum_{k=0}^{K-1} B_k\right] \tag{21}$$

Among them, if $\lambda$ is the Lagrange multiplier, and the derivative of $f_\lambda$ concerning $B_k$, $C_k$, and $\lambda$ is 0, we can obtain

$$B_k = -\frac{\lambda}{2}\frac{|H_k|^2 + \sigma_\omega^2}{\sigma_\omega^2} - 1 \tag{22}$$

$$C_k = \frac{(B_k + 1)H_k^*}{|H_k|^2 + \sigma_\omega^2} \tag{23}$$

$$\lambda = \frac{-2\sigma_\omega{}^2}{\frac{1}{P}\sum\limits_{p=0}^{P-1}\left(|H_p|^2 + \sigma_\omega{}^2\right)} \tag{24}$$

It can be seen from the above conclusions that the calculation of the feedforward filter coefficients and feedback filter coefficients in the MMSE-IBDFE algorithm is no longer related to the correlation factor, thus eliminating the influence of the correlation factor estimation error on the equalization accuracy. In addition, the computational complexity and complexity of the algorithm are also reduced.

4.2.2. Estimated $\delta$-Based Iterative Block Decision Feedback Equalization

It can be seen from reference [19] that the filter coefficient in the MMSE equalization is $W_k = \frac{H_k{}^*}{|H_k|^2 + \frac{\sigma_\omega{}^2}{\sigma_x{}^2}}$. In order to facilitate the analysis, the power of the data signal is normalized, that is, assuming $\sigma_x{}^2 = 1$, the obtained expression is

$$W_k = \frac{H_k{}^*}{|H_k|^2 + \sigma_\omega{}^2} \tag{25}$$

From Equation (2), the data after MMSE equalization can be obtained as

$$\begin{aligned}
\tilde{X}_k &= W_k Y_k \\
&= X_k - \frac{X_k \sigma_\omega{}^2}{|H_k|^2 + \sigma_\omega{}^2} + \frac{H_k{}^* V_k}{|H_k|^2 + \sigma_\omega{}^2} \\
&= X_k + \Delta_k + \tilde{V}_k
\end{aligned} \tag{26}$$

where

$$\Delta_k = -\frac{X_k \sigma_\omega{}^2}{|H_k|^2 + \sigma_\omega{}^2} \tag{27}$$

$$\tilde{V}_k = \frac{H_k{}^* V_k}{|H_k|^2 + \sigma_\omega{}^2} \tag{28}$$

The time domain expression of Equation (26) is

$$\tilde{x}_n = x_n + \delta_n + \tilde{v}_n \tag{29}$$

In Equation (29), $\delta_n$ and $\tilde{v}_n$, respectively, represent the RISI and noise interference after MMSE equalization, and MMSE equalization directly makes a decision on $\tilde{x}_n$, ignoring the influence of RISI and noise, resulting in inaccurate equalization data. Because the MMSE equalization has good performance in the deep fading pole of the channel frequency domain and is less affected by the noise, the elimination improvement is carried out for RISI.

After the improvement, the principle block diagram of the E-IBDFE algorithm is shown in Figure 5.

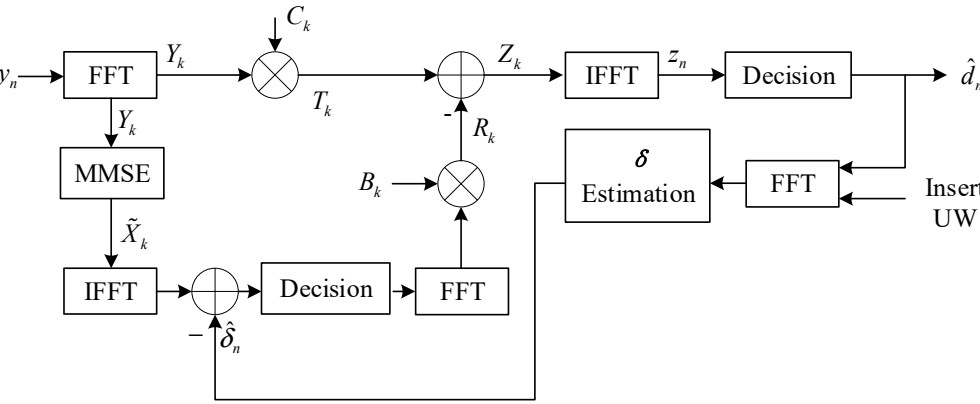

**Figure 5.** Schematic diagram of E-IBDFE.

It can be seen from Figure 5 that in this paper, $\delta$ estimation is added before the decision in the MMSE equalization branch to eliminate RISI. The algorithm steps of estimation are as follows.

① Insert the decision signal $\hat{d}_n$ into UW, and then perform FFT to obtain $\hat{D}_k$.

② Substituting $\hat{D}_k$ for $X_k$ into Equation (27), the estimated value $\hat{\Delta}_k$ of $\Delta_k$ is obtained, and the expression of $\hat{\Delta}_k$ is

$$\hat{\Delta}_k = -\frac{\hat{D}_k \sigma_\omega{}^2}{|H_k|^2 + \sigma_\omega{}^2} \tag{30}$$

③ Using $\hat{\Delta}_k$ to calculate the estimated value $\hat{\delta}_n$ of $\delta_n$, we obtain

$$\hat{\delta}_n = IFFT\left\{\hat{\Delta}_k\right\} \tag{31}$$

④ Eliminate $\hat{\delta}_n$ from $\tilde{X}_k$ and make a decision on $\left(\tilde{X}_k - \hat{\delta}_n\right)$.

The above algorithm uses $\hat{D}_k$ instead of $X_k$ to estimate $\delta_n$ in step ②, and the error is smaller than RISI when the bit error rate is low, so the decision accuracy can be improved. The algorithm can also be iterative, but it will increase the amount of calculation, so whether to iterate or not can be considered according to the accuracy requirements.

Compared with the classical IBDFE algorithm, the MMSE-IBDFE algorithm greatly reduces the complexity of the operation due to the cancellation of the iterative mechanism. The complexity of the E-IBDFE equalization algorithm is slightly higher than the former but is still lower than the classical IBDFE algorithm.

The complexity comparison between the three algorithms and the MMSE equalization algorithm is shown in Table 1, where $N$ denotes the number of FFT conversion points and $N_l$ denotes the number of feedforward filter coefficients and feedback filter coefficients iterations of the IBDFE algorithm. As can be seen in Table 1, the operational complexity of the IBDFE algorithm is directly related to the number of iterations $N_l$, and several iterations are required to achieve high judgment accuracy. The complexity of the E-IBDFE algorithm is only related to the number of data per frame, and when $N_l \geq 4$, the complexity of the operation and the computation of the coefficients are smaller than those of the IBDFE algorithm.

**Table 1.** Comparison of complexity.

| Algorithm Category | Algorithmic Complexity | Coefficient Calculation Amount |
|---|---|---|
| MMSE | $N \log_2 N + N$ | $N$ |
| IBDFE | $(2N_l + 1) \frac{N}{2} \log_2 N + 2N_l N$ | $(4N_l - 2)N$ |
| MMSE-IBDFE | $2N \log_2 N + 2N$ | $3N$ |
| E-IBDFE | $3N \log_2 N + 2N$ | $4N$ |

## 5. Simulations and Results

To verify the effectiveness of our E-IBDFE algorithm, several simulations are conducted. The simulation platform is MATLAB with version R2018a, and according to reference [26], the channel model adopts the Extended Vehicular A (EVA) channel, and the channel-related parameter settings are shown in Table 2. The communication model adopts the SC-FDE system, assuming that the channel synchronization and channel estimation are ideal, and the related parameters of the system are set as shown in Table 3.

**Table 2.** Parameters of channel EVA.

| Path | Delay/μs | Normalized Power |
|---|---|---|
| 1 | 0 | 0.249 |
| 2 | 0.03 | 0.432 |
| 3 | 0.15 | 0.731 |
| 4 | 0.31 | 0.615 |
| 5 | 0.37 | 0.818 |
| 6 | 0.71 | 0.534 |
| 7 | 0.109 | 0.478 |
| 8 | 0.173 | 0.173 |

**Table 3.** Simulation parameters of SC-FDE system.

| System Parameters | Value |
|---|---|
| Modulation | QPSK |
| FFT length | 256 |
| UW type | Chu array |
| UW length | 32 |
| Maximum Doppler shift | 20 Hz |
| Symbol period | 0.2 μs |
| Channel coding | None |
| Constellation mapping | Gray Code |

Figure 6 shows the BER performance simulation results of the E-IBDFE algorithm and the classical IBDFE algorithm with two iterations under EVA channel conditions. As can be seen from the figure, the performance of the classical IBDFE algorithm increases with the increase in the number of iterations, but the computational complexity also increases with the increase in the number of iterations. When the bit error rate is $10^{-2}$, the signal-to-noise ratio (SNR) of the E-IBDFE algorithm is about 1 dB higher than that of the classical IBDFE algorithm after two iterations. This is because ZF equalization forces ISI to zero but does not take into account the noise amplification phenomenon, which is improved by the E-IBDFE algorithm to enhance the equalization performance. However, by increasing the number of iterations to further eliminate the error and to compensate for the effect of noise interference, the IBDFE algorithm with four iterations has a gain of about 1.6 dB in SNR compared with the E-IBDFE algorithm when the bit error rate is $10^{-2}$, but its computational complexity is much higher than that of the E-IBDFE algorithm.

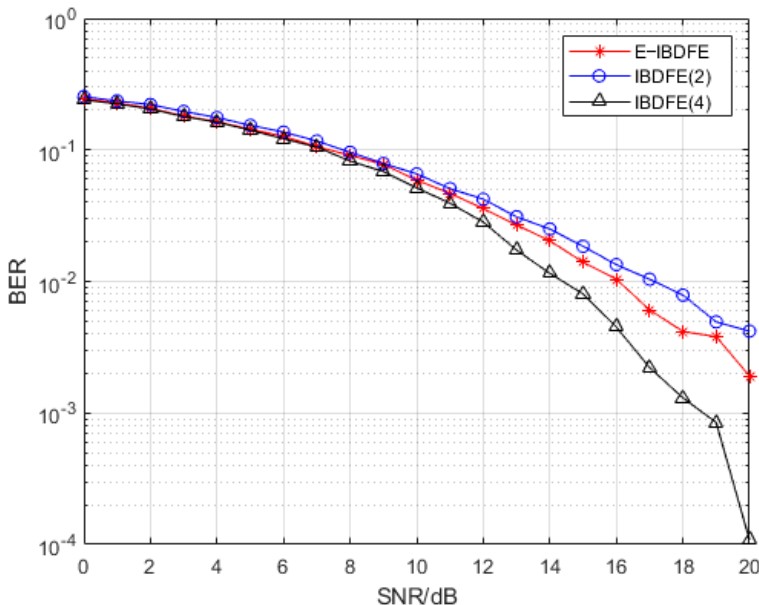

**Figure 6.** Comparison of BER of E−IBDFE and IBDFE in EVA channel.

Figure 7 shows the BER performance results of the E-IBDFE algorithm in the additive white Gaussian noise (AWGN) environment and the Middleton-A noise environment. It can be seen that the algorithm has good performance in both Gaussian and non-Gaussian noise environments.

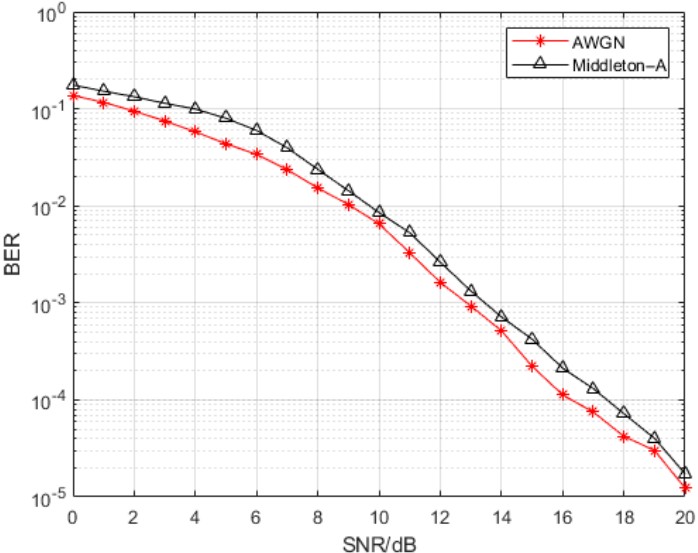

**Figure 7.** Comparison of BER of E−IBDFE in AWGN and Middleton−A noise environments.

Figure 8 is a simulation comparison of four algorithms, the MMSE equalization algorithm, IBDFE algorithm, MMSE-IBDFE algorithm, and E-IBDFE algorithm, under the condition of the EVA channel. As can be seen from the figure, the performance of the MMSE equalization algorithm is the worst because there is still a strong RISI after MMSE equalization. In the case of lower SNR, the performances of the MMSE and MMSE-IBDFE algorithms are poor because the low SNR leads to the increase in noise interference and additional interference caused by RISI estimation deviation, and the error further accumulates, which deteriorates the system's performance. The algorithm proposed in this paper is improved for the above problems, so the corresponding interference is reduced and the equalization performance is improved. In addition, with the increasing SNR, ISI will become an important influence factor on the system equalization performance, and the

BER performance gap of each equalization method will gradually become larger, which indicates that the advantage of E-IBDFE equalization is more obvious under the high SNR condition.

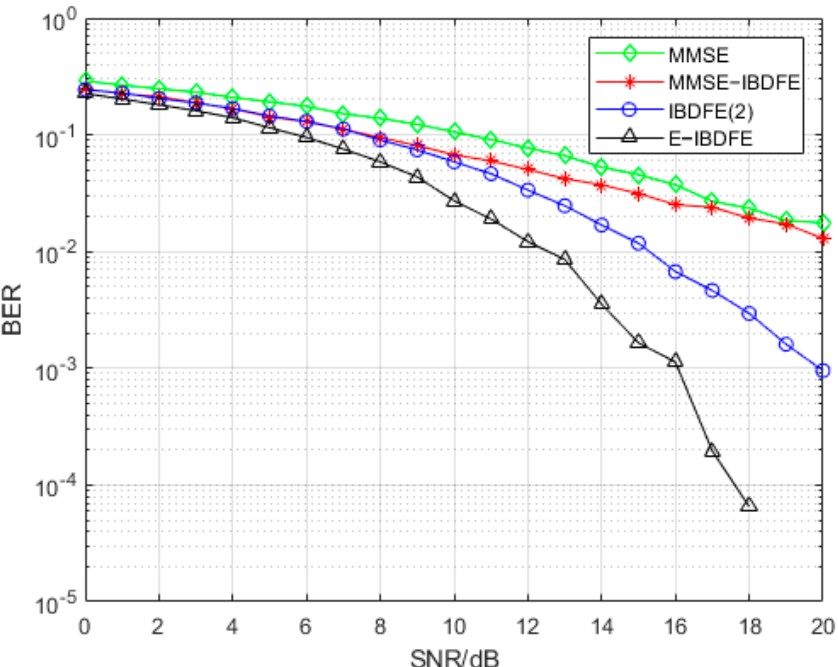

**Figure 8.** Comparison of BER of 4 equalization algorithms in EVA channel.

## 6. Conclusions

In this paper, the classical nonlinear equalization algorithm in the SC-FDE system, the IBDFE algorithm, is studied. Aiming at the shortcomings of the classical IBDFE algorithm where the correlation factor needs to be updated in each iteration, and the computational complexity of the algorithm increases greatly with the increase in the number of iterations, the feedback branch is improved. Using MMSE equalization instead of ZF equalization, canceling iteration, and adding $\delta$ estimation to eliminate RISI of the feedback branch, the estimated $\delta$-based iterative block decision feedback equalization algorithm is proposed and the complexity of the algorithm is analyzed. The simulation results show that the E-IBDFE algorithm has better equalization performance than the IBDFE algorithm with lower iteration times, the E-IBDFE algorithm has lower computational complexity when the performance is similar, and the E-IBDFE algorithm has better performance in different noise environments. In future research, we will combine the SC-FDE system with spatial diversity and spatial multiplexing techniques to extend it to MIMO channels, and then study the application of frequency domain equalization algorithms in the MIMO-SCFDE system.

**Author Contributions:** Writing—original draft, Y.L.; Writing—review & editing, X.C., D.Y. and D.H. All authors have read and agreed to the published version of the manuscript.

**Funding:** This research received no funding.

**Data Availability Statement:** The data used to support the findings of this study are included within the article.

**Acknowledgments:** I thank my teachers, friends, and other colleagues for their discussions on simulation and comments on this paper.

**Conflicts of Interest:** The authors declare that there are no conflict of interest regarding the publication of this paper.

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
