# Peer review of "An Estimated δ-Based Iterative Block Decision Feedback Equalization in SC-FDE System"

_electronics, doi:10.3390/electronics11203397_

Round 1

Reviewer 1 Report

Altough the paper deals with an interesting research topic, I have some concerns prior publication:

1) Litterature review should be expanded. Moreover, the key novelty of the present work is unclear.

2) There are many equations in the paper however it is hard for the reader to follow since no references are provided.

3) Simulation results section should be expanded with more details on the overall setup. For example, how many active users do you consider? What was your simulation platform?

4) In the same context, it seems that a BER in the order of 10^(-3) is achieved for very high SNR values. Please elaborate on this.

5) Litterature review should be expanded with more recent works. 

Author Response

请参阅附件。

Reviewer 2 Report

1. Abstract needs to be enhanced with giving information about the work proposed, implementation, results obtained and improvements.

2. Keywords are not in the standard format. It has to be updated.

3. Related work must be included as a separate section and provide more information of the problem definition and objectives.

4. Figure 1 name is not appropriate, kindly update it.

5. Equation numbering is not aligned uniformly, kindly update it.

6. Include good quality figures/graphics, Tables must be drawn in table format, not as an image.

7. Improve the Experimental Results. It would be better if some more comprehensive evaluations are included.

8. The simulation results must be detailed properly and inference must be drawn from the same.

9. Authors must highlight the novelty and main contributions clearly

10. Future enhancement can be included in the conclusion section.

Round 2

Reviewer 1 Report

The authors have revised their work according to my suggestions, therefore it can be published in its current form.

Reviewer 2 Report

The paper has been updated according to the review comments.